# Selection of GalNAc-conjugated siRNAs with limited off-target-driven rat hepatotoxicity

Maja M. Janas[1], Mark K. Schlegel[1], Carole E. Harbison[1], Vedat O. Yilmaz[1], Yongfeng Jiang[1], Rubina Parmar[1], Ivan Zlatev [1], Adam Castoreno[1], Huilei Xu[1], Svetlana Shulga-Morskaya[1], Kallanthottathil G. Rajeev[1], Muthiah Manoharan[1], Natalie D. Keirstead[1], Martin A. Maier[1] & Vasant Jadhav[1]

Small interfering RNAs (siRNAs) conjugated to a trivalent *N*-acetylgalactosamine (GalNAc) ligand are being evaluated in investigational clinical studies for a variety of indications. The typical development candidate selection process includes evaluation of the most active compounds for toxicity in rats at pharmacologically exaggerated doses. The subset of GalNAc-siRNAs that show rat hepatotoxicity is not advanced to clinical development. Potential mechanisms of hepatotoxicity can be associated with the intracellular accumulation of oligonucleotides and their metabolites, RNA interference (RNAi)-mediated hybridization-based off-target effects, and/or perturbation of endogenous RNAi pathways. Here we show that rodent hepatotoxicity observed at supratherapeutic exposures can be largely attributed to RNAi-mediated off-target effects, but not chemical modifications or the perturbation of RNAi pathways. Furthermore, these off-target effects can be mitigated by modulating seed-pairing using a thermally destabilizing chemical modification, which significantly improves the safety profile of a GalNAc-siRNA in rat and may minimize the occurrence of hepatotoxic siRNAs across species.

[1] Alnylam Pharmaceuticals, 300 Third Street, Cambridge, MA 02142, USA. Maja M. Janas and Mark K. Schlegel contributed equally to this work. Correspondence and requests for materials should be addressed to V.J. (email: vjadhav@alnylam.com)

RNA interference (RNAi) is a highly conserved silencing pathway whereby short double-stranded RNAs down-regulate expression of complementary target mRNAs to fine-tune gene expression and to protect against invasive nucleic acids[1]. These 20–25-nucleotide-long duplexes load into the RNA-induced silencing complex (RISC), which retains the antisense (guide) strand and discards the sense (passenger) strand[2]. There are two types of RNA-silencing triggers that can be loaded into cytoplasmic RISC in somatic cells: microRNAs (miRNAs) and siRNAs. miRNAs recognize target mRNAs via a partial sequence match predominantly within the 3′ untranslated region (3′UTR) leading to translational repression and mRNA destabilization, while siRNAs recognize target mRNAs via a full sequence match leading to mRNA cleavage[1]. Because of differences in the mechanism of action, siRNA-mediated gene-silencing activity is more potent than miRNA-mediated gene silencing. The core component of RISC is a member of the Argonaute (Ago) protein superfamily[3]. Although only Ago2 can catalyze siRNA-mediated mRNA cleavage[4], all four Ago proteins (Ago1–4) participate in miRNA-mediated silencing, the major endogenous RNAi pathway in mammals.

This potent and selective natural RNAi pathway can be harnessed by exogenous siRNAs to knock down disease-causing mRNAs[5]. The liver is an attractive target organ for RNAi therapeutics owing to its association with a large number of human diseases combined with the possibility for efficient delivery. Conjugation to a trivalent N-acetylgalactosamine (GalNAc) ligand allows targeted delivery to hepatocytes via the abundant, hepatocyte-specific, and rapidly recycling asialoglycoprotein receptor[6–8]. To confer optimal drug-like properties (pharmacokinetics, pharmacodynamics, safety), we utilize siRNAs which are chemically modified. Typical modifications of the 2′-position of the ribose moiety include 2′-O-methyl (2′OMe) and 2′-deoxy-2′-fluoro (2′F), which, when combined with phosphorothioate (PS) modification at certain positions in the phosphodiester backbone[8–11], have previously been reported to be well-tolerated in siRNAs in vitro[8, 12, 13].

Our process of selecting GalNAc-siRNA conjugates for clinical development includes screening the most potent set of compounds in short-term rat toxicity studies at pharmacologically exaggerated exposures in a repeat-dose regimen. On average, we observe an approximate 40% attrition rate in these toxicity screens. We are generally able to identify both non-toxic and toxic siRNA sequences for the same target mRNA, arguing against on-target toxicity. The typical pathology findings that result in compound attrition are hepatocellular degeneration and necrosis with clinical pathology changes such as elevated alanine aminotransferase, aspartate aminotransferase, and total bilirubin. GalNAc-siRNAs that have an acceptable safety profile in these rat studies advance towards development candidate selection.

There are several potential mechanisms of hepatotoxicity to consider based upon the uptake pathway and pharmacologic activity of GalNAc-siRNAs. Mechanisms upstream of RISC loading could include class-wide toxicities associated with the accumulation of the endocytosed material in endosomes and lysosomes, or toxicities associated with chemical modifications of siRNAs or their metabolites. Endo-lysosomal accumulation of metabolically stable compounds could result in perturbation of endogenous endocytic processes by altering the structure and/or function of endocytic vesicles or proteins. Chemistry-related toxicities could result from non-specific binding to cellular proteins mediated by the chemical modifications used to protect siRNAs against nuclease digestion and immune recognition[9, 10], leading to altered protein localization, structure, or function[14, 15]. Indeed, chemistry-related toxicities are sometimes observed with heavily modified oligonucleotide therapeutics both in vitro and in vivo. While this is especially true for single-stranded oligonucleotides with a high PS content that exhibit strong, non-sequence-specific protein binding[14–16], these effects have not been reported to date with low PS content siRNAs[12, 17].

Hepatotoxicity may also result from mechanisms downstream of RISC loading, including competition with the endogenous RNAi pathways and/or off-target repression of partially complementary mRNAs in a miRNA-like fashion. Since mammalian RISC is naturally occupied by miRNAs, it is conceivable that siRNAs could compete with miRNAs for RISC loading, resulting in miRNA destabilization and de-repression of their endogenous target mRNAs. Indeed, previous reports suggested that the major driver of small hairpin RNA hepatotoxicity in mice is perturbation of the most abundant hepatocyte-specific miRNA, miR-122[18, 19]. As described previously, siRNA-loaded RISC or antisense oligonucleotides may also bind and repress mRNAs with partial sequence complementarity[20–24]. Such RNAi-mediated off-target effects mimic the weak post-transcriptional silencing by endogenous miRNAs and are primarily driven by binding of the antisense seed region (nucleotides 2–8) to complementary sites in the 3′UTR[25].

Here we describe a series of mechanistic studies demonstrating that RNAi-mediated, seed hybridization-based off-target effects are the major driver of hepatotoxicity observed with GalNAc-siRNAs in rodent toxicity screens. We find that blocking RISC loading of a random subset of hepatotoxic GalNAc-siRNAs without altering the PS, 2′OMe, or 2′F content mitigates hepatotoxicity while maintaining siRNA liver exposure, indicating little to no contribution of siRNA chemical modifications. Furthermore, blocking activity of the RISC-loaded antisense strand with complementary short oligonucleotides[26], changing the sequence of the seed region without altering the PS, 2′OMe, or 2′F content, or utilizing a destabilizing glycol nucleic acid (GNA) nucleotide in the seed region also mitigate hepatotoxicity while maintaining comparable siRNA levels in both total liver and RISC. These results indicate that RISC loading is necessary but not sufficient for hepatotoxicity. Consistent with these data, global transcriptome profiling by RNAseq shows RISC loading-dependent downregulation of RNAs enriched for a seed match to the antisense but not the sense strand of GalNAc-siRNAs. Taken together, these studies implicate sequence-specific RNAi-based off-target effects, not a class effect based on siRNA chemistry or competition for RISC loading[27], as the major driver of hepatotoxicity in rodents.

## Results

**Blocking siRNA RISC loading mitigates hepatotoxicity.** Efficient RISC loading and activity of small RNAi triggers depends on the presence of a monophosphate moiety at the 5′-end[28, 29]. While endogenous miRNAs naturally contain a 5′-monophosphate as a result of their biogenesis[1], exogenous siRNAs are thought to be dependent on phosphorylation by kinases following intracellular uptake[30, 31]. To characterize the relationship of RISC loading to the hepatotoxicity observed with a random subset of modified GalNAc-siRNAs (Supplementary Fig. 1)[8] in rodent toxicity studies, the 5′-end of the duplex antisense strand with previously established hepatotoxicity were capped (Fig. 1a) using three different types of nucleotide modifications designed to impede 5′-phosphorylation and thus RISC loading: 5′-inverted abasic (iB)[32], 5′-deoxy-5′-(4-morpholinyl), or 5′-deoxy nucleotides (Parmar et al., manuscript in preparation). These capped siRNAs defective in RISC loading had the same PS, 2′OMe, and 2′F content as their RNAi-active counterparts that were identified in previous short-term repeat-dose rat toxicity screening studies as hepatotoxic and were designed against various target mRNAs with or without expected on-target activity in rodents.

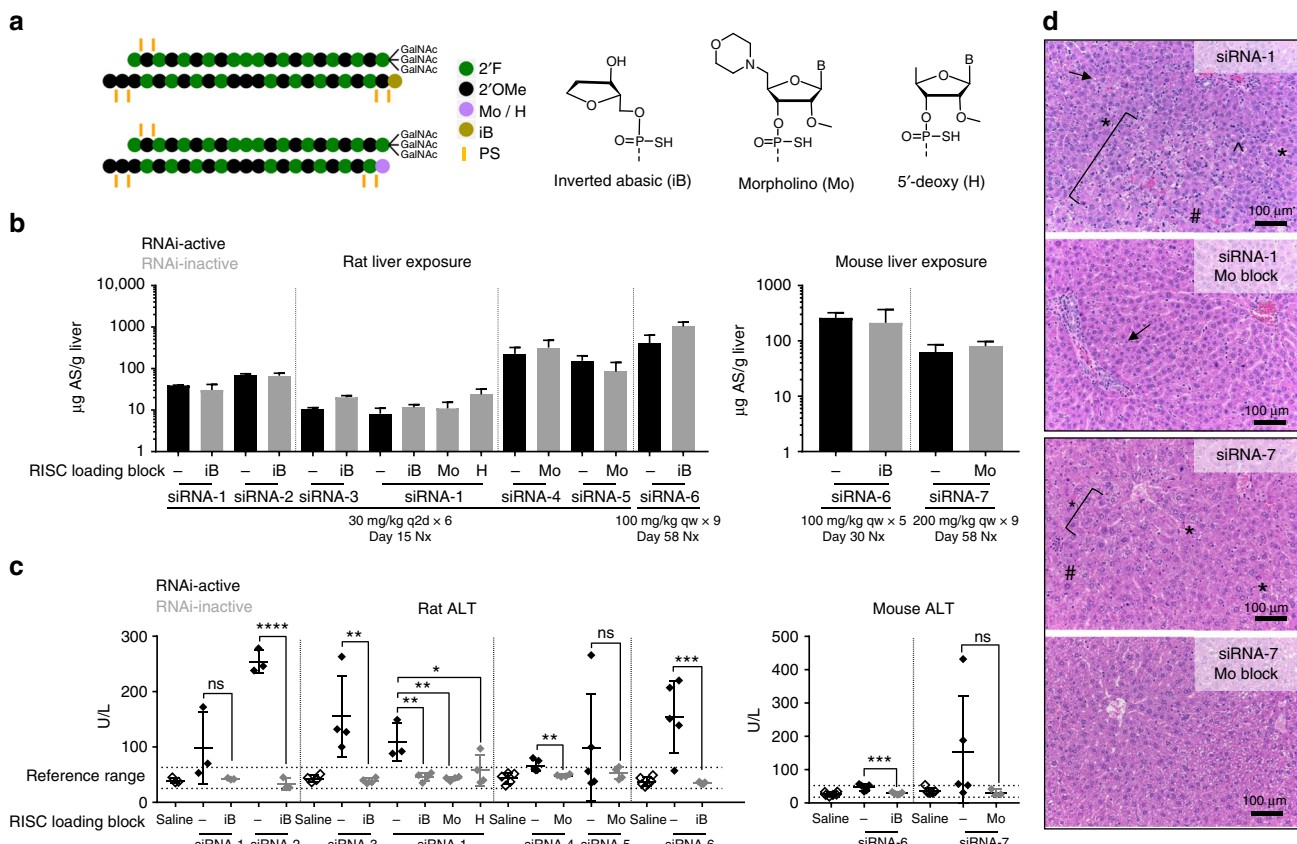

**Fig. 1** Blocking RISC loading mitigates hepatotoxicity. **a** Structures of nucleotide analogs used at 5′-ends of siRNAs to prevent 5′-phosphorylation thus reducing RISC loading. **b** Liver exposures for parent (RNAi-active) and capped (RNAi-inactive) GalNAc-siRNAs in rat and mouse toxicity studies as assessed by stem-loop RT-qPCR for the antisense strand (AS) at necropsy (nx). Dashed vertical lines demarcate studies conducted separately. **c** Serum alanine aminotransferase (ALT) levels measured at necropsy. Differences between group means were evaluated for statistical significance using one way ANOVA with post hoc corrections (for multiple siRNAs) in GraphPad Prism 7. ns, not significant; *$p < 0.05$; **$p < 0.01$; ***$p < 0.001$; ****$p < 0.0001$. Error bars represent standard deviation of the mean. **d** H&E staining of liver sections collected at necropsy. In the rat, hepatotoxic siRNAs (siRNA-1 shown here) had hepatocellular degeneration (bracketed area), increased sinusoidal cells due to Kupffer cell hyperplasia and/or leukocyte infiltration (#), single cell necrosis (*), increased mitoses (^), and hepatocellular vacuolation (arrow). In the mouse, hepatotoxic siRNAs (siRNA-7 shown here) were associated with single-cell necrosis and lower incidence and severity of the other findings commonly seen in the rat. Capped RNAi-inactive siRNAs had minimal vacuolation or no histologic findings in both species. Cytoplasmic clearing present in the mice was consistent with glycogen due to incomplete fasting and was not considered test article-related. Microscopic liver findings for all tested siRNAs are tabulated in Supplementary Table 1. $N = 3$ males (6–8 weeks old) per group; qw, weekly dosing; q2d, every other day dosing

The effects of blocking RISC loading on hepatotoxicity were tested at toxicological doses in rodents. Rats or mice received 5–9 weekly or every other day doses of 30–100 mg/kg, which represented 2–3 log exaggeration of the pharmacological dose range. As expected, phosphorylation-blocking 5′-capping modifications of the antisense strand reduced RISC loading (Supplementary Fig. 2a) and target mRNA knockdown (Supplementary Fig. 2b) relative to parent siRNAs. Across all studies, there were no significant differences in liver concentrations between RNAi-active and RNAi-inactive siRNAs of the same sequence and backbone chemistry (Fig. 1b), confirming that the endo-lysosomal system and intracellular proteins were exposed to equivalent amounts of each siRNA regardless of its RISC-loading ability. Despite equivalent liver exposures, blocking RISC loading of known hepatotoxic siRNAs eliminated liver enzyme elevations (Fig. 1c; Supplementary Fig. 2c) and most to all microscopic liver findings, including fibrosis, single cell necrosis, and hepatocellular degeneration in both mice and rats (Fig. 1d; Supplementary Table 1). Importantly, placing modifications which block RISC loading on the 5′-end of the sense strand alone (Supplementary Fig. 3a-c) or on a non-toxic toolkit GalNAc-siRNA

(Supplementary Fig. 4a-c) had no effects on liver enzyme elevations or microscopic liver findings (Supplementary Tables 2 and 3), indicating that these 5′-caps are unlikely to impact intracellular trafficking of siRNAs or introduce additional safety liabilities. These studies indicate that rodent hepatotoxicity of a subset of GalNAc-siRNAs is dependent on RISC loading of the antisense strand but independent of siRNA chemistry-related mechanisms upstream of RISC loading, such as perturbation of the endo-lysosomal system or undesired intracellular protein binding to the relatively hydrophobic backbone modifications such as PS or 2′F.

**Altering siRNA chemistry does not mitigate hepatotoxicity.** In order to further de-risk the potential contribution of 2′F and 2′OMe content to siRNA hepatotoxicity, two versions of a model hepatotoxic siRNA with differing patterns of chemical modification were tested in rodent toxicity studies: a high 2′F version (48% 2′F and 52% 2′OMe) and a low 2′F version (21% 2′F and 79% 2′OMe) (Fig. 2a)[11]. Both compounds had identical sequence and PS content and retained potent silencing activity

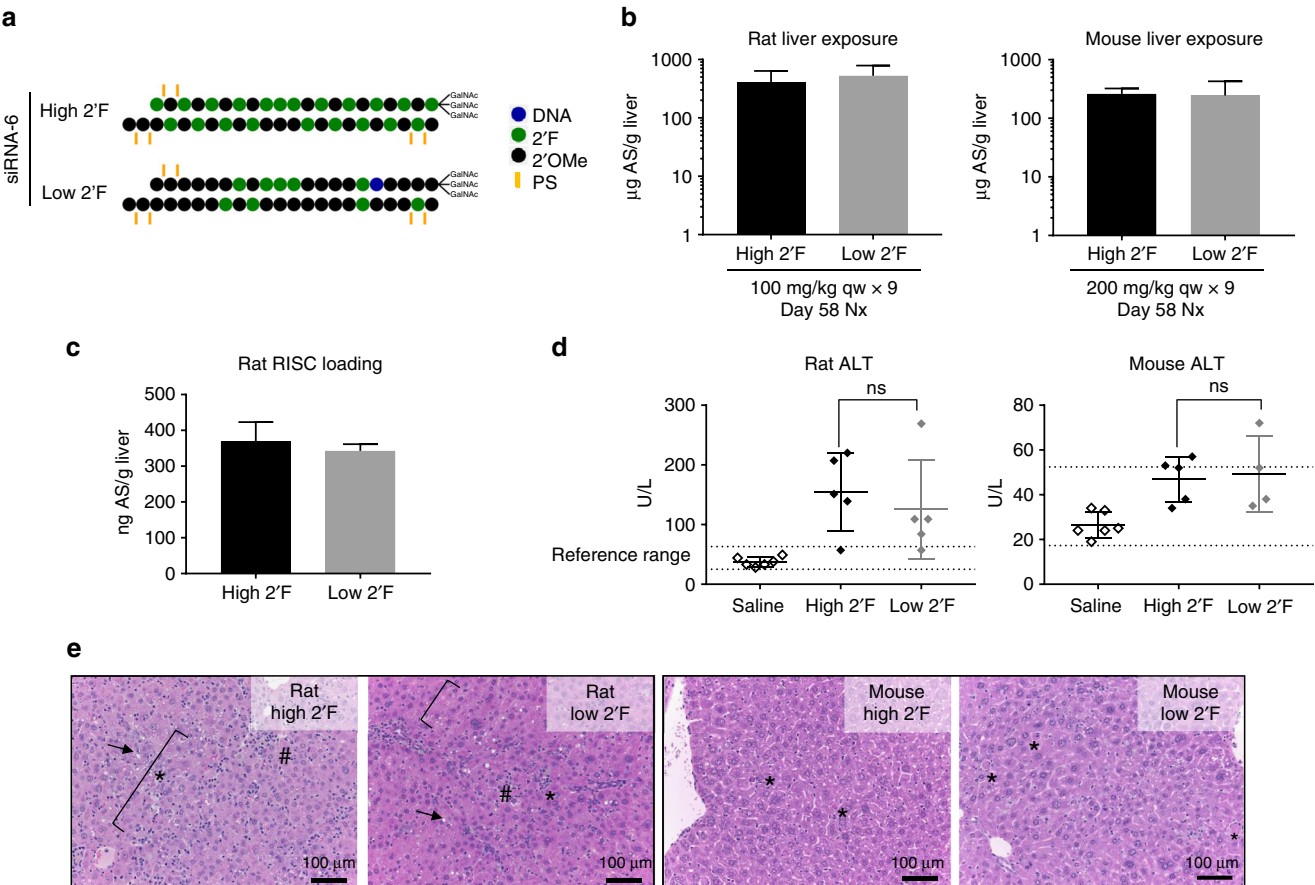

**Fig. 2** Changing siRNA chemical modifications does not mitigate hepatotoxicity. **a** Chemical modification patterns of the high 2′F and low 2′F GalNAc-siRNAs with the same PS content and sequence. **b** Liver exposures in rat and mouse toxicity studies as assessed by stem-loop RT-qPCR for the antisense strand (AS) at necropsy (nx). **c** Liver RISC loading as assessed by stem-loop RT-qPCR for the antisense at necropsy. **d** Serum alanine aminotransferase (ALT) levels measured at necropsy. Differences between group means were evaluated for statistical significance using one way ANOVA with post hoc corrections (for multiple siRNAs) in GraphPad Prism 7. ns, not significant. Error bars represent standard deviation of the mean. **e** H&E staining of liver sections collected at necropsy. In the rat, both high 2′F and low 2′F siRNA-6 compounds were associated with hepatocellular degeneration (bracket), single cell necrosis (*), increased sinusoidal cells consistent with Kupffer cell hyperplasia and/or infiltrating leukocytes (#), and hepatocellular vacuolation (arrow). In the mouse, findings consisted of single-cell necrosis for both chemical modification patterns. All microscopic liver findings are tabulated in Supplementary Table 4. N = 3 males (6–8 weeks old) per group; qw weekly dosing

(Supplementary Fig. 5). These compounds were dosed weekly in rats at 100 mg/kg and in mice at 200 mg/kg over the course of 9 weeks. With this frequent dosing paradigm, liver exposures (Fig. 2b) and RISC loading (Fig. 2c) were comparable for the low and high 2′F siRNAs at the end of each study. Similarly, liver enzyme elevations (Fig. 2d) and microscopic liver findings (Fig. 2e; Supplementary Table 4) were independent of the number of 2′F or 2′OMe modifications in this sequence in both rodent species. These data provide further evidence against siRNA chemical modifications as the driving force behind rodent hepatotoxicity of GalNAc-siRNAs.

**Blocking siRNA-loaded RISC activity mitigates hepatotoxicity.**
Since siRNA chemistry-related mechanisms upstream of RISC loading did not appear to have a significant impact on hepatotoxicity in rodents, we focused on distinguishing RNAi-mediated off-target effects from the perturbation of endogenous RNAi pathways. The strategy allowed for siRNA RISC loading by keeping the siRNA chemistry and sequence unchanged, but prevented binding of siRNA-loaded RISC to potential off-target mRNAs. To achieve this, we blocked RNAi activity downstream of RISC loading using GalNAc-conjugated, short single-stranded

oligonucleotides complementary to the siRNA antisense strand, known as REVERSIR[TM] compounds[26, 33], in two types of rat toxicity studies: prevention and treatment (Fig. 3a).

In prevention studies, REVERSIR[TM] molecules complementary to the antisense strand of a hepatotoxic siRNA or a control scrambled REVERSIR[TM] sequence of the same length and chemistry composition were pre-dosed at high pharmacological doses (3 or 10 mg/kg) either 24 h before the first siRNA dose or 24 h before the first and second siRNA dose. In treatment studies, REVERSIR[TM] compounds were dosed at high pharmacological doses (3 or 10 mg/kg) 24 h after the last siRNA dose. Hepatotoxic GalNAc-siRNAs were dosed weekly (three times) or every other day (six times) at 30 mg/kg. Both the complementary and the scrambled REVERSIR[TM] molecules were confirmed bioinformatically to exhibit no full complementarity to any liver-expressed miRNAs that could potentially be blocked by REVERSIR[TM] compounds.

REVERSIR[TM] treatment pre-siRNA or post-siRNA administration reduced on-target knockdown (Supplementary Fig. 6) but did not affect liver siRNA levels (Fig. 3b) or RISC loading (Fig. 3c). However, the complementary REVERSIR[TM] compounds (RVR-1, RVR-4, or RVR-5) but not the control, scrambled REVERSIR[TM] (Ctr RVR) reduced the liver enzyme

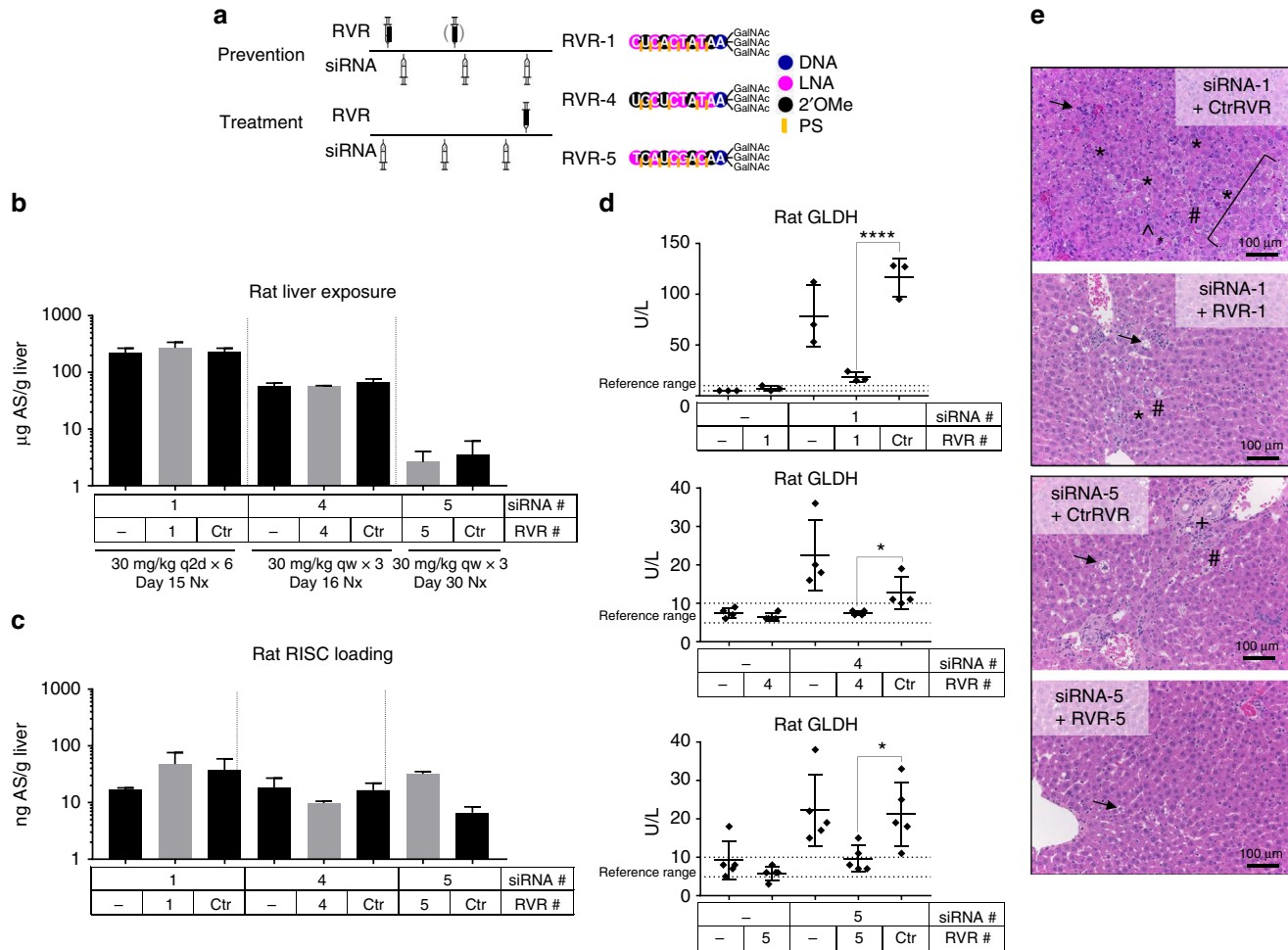

**Fig. 3** Blocking antisense strand-loaded RISC activity mitigates hepatotoxicity. **a** Study design depicting prevention and treatment of rat toxicity by GalNAc-siRNAs using REVERSIR[TM]. **b** Liver exposures for GalNAc-siRNAs in rat prevention (siRNA-1 and siRNA-4) or treatment (siRNA-5) toxicity studies as assessed by stem-loop RT-qPCR for the antisense strand (AS) at necropsy (nx). **c** Liver RISC loading with or without REVERSIR[TM] treatment as assessed by stem-loop RT-qPCR for the antisense strand at necropsy. **d** Serum glutamate dehydrogenase (GLDH) levels measured at necropsy. Differences between group means were evaluated for statistical significance using one way ANOVA with post hoc corrections (for multiple siRNAs) in GraphPad Prism 7. ns. not significant; *$p < 0.05$; ****$p < 0.0001$. Error bars represent standard deviation of the mean. **e** H&E staining of liver sections collected at necropsy. Known toxic siRNAs administered alone or with a scrambled, control (Ctr) REVERSIR[TM] were associated with hepatocellular degeneration (bracket), single cell necrosis (*), increased sinusoidal cells consistent with Kupffer cell hyperplasia and/or infiltrating leukocytes (#), increased mitoses (^), bile duct hyperplasia with fibrosis (+), and hepatocellular vacuolation (arrow). Co-administration of a complementary REVERSIR[TM] decreased the severity of these findings and often limited their distribution. All microscopic liver findings are tabulated in Supplementary Table 5. $N = 3$ males (6–8 weeks old) per group; qw, weekly dosing; q2d, every other day dosing

elevations observed with their respective targets, siRNA-1, siRNA-4, or siRNA-5 (Fig. 3d), and decreased the severity and incidence of microscopic liver findings (Fig. 3e; Supplementary Table 5). REVERSIR[TM] compounds administered alone had no toxic effects (Fig. 3d). By deploying the REVERSIR[TM] approach, siRNA-induced hepatotoxicity was mitigated without affecting RISC loading and without changing siRNA chemistry. Thus, these data support the hypothesis that hepatotoxicity is driven by antisense strand-mediated RNAi off-target effects, and not by competition for RISC complexes with endogenous RNAi pathways[27] or siRNA chemistry-mediated effects.

**Swapping seed regions mitigates hepatotoxicity.** Analogous to miRNA mechanisms, RNAi-mediated off-target effects of siRNAs are typically driven by the seed region of the guide strand[20–22]. If these effects elicit the observed rodent hepatotoxicity of GalNAc-siRNAs, the sequence of the seed region and not the flanking region outside nucleotides 2–8 should be a key determining factor of whether a specific sequence is associated with hepatotoxicity or not. To test this hypothesis, the seed region of a hepatotoxic siRNA was replaced with the seed region of a non-hepatotoxic siRNA, while the seed region of a non-hepatotoxic siRNA was replaced with the seed region of a hepatotoxic siRNA, without changing the chemical modification pattern (Fig. 4a).

The two seed-swapped siRNAs along with the parent hepatotoxic and non-hepatotoxic siRNAs were administered to rats at a toxicological dose of 30 mg/kg six times every other day. Liver exposures were comparable for all four compounds (Fig. 4b). RISC loading was lower for the toxic parent siRNA as well as the siRNA containing the toxic seed region relative to the non-toxic parent siRNA or the siRNA containing the non-toxic seed region (Fig. 4c). Despite the lower levels of RISC loading, however, the two siRNAs containing a toxic seed region were most hepatotoxic, arguing against competition for RISC loading as the major driver of hepatotoxicity[27]. Replacing a toxic seed region

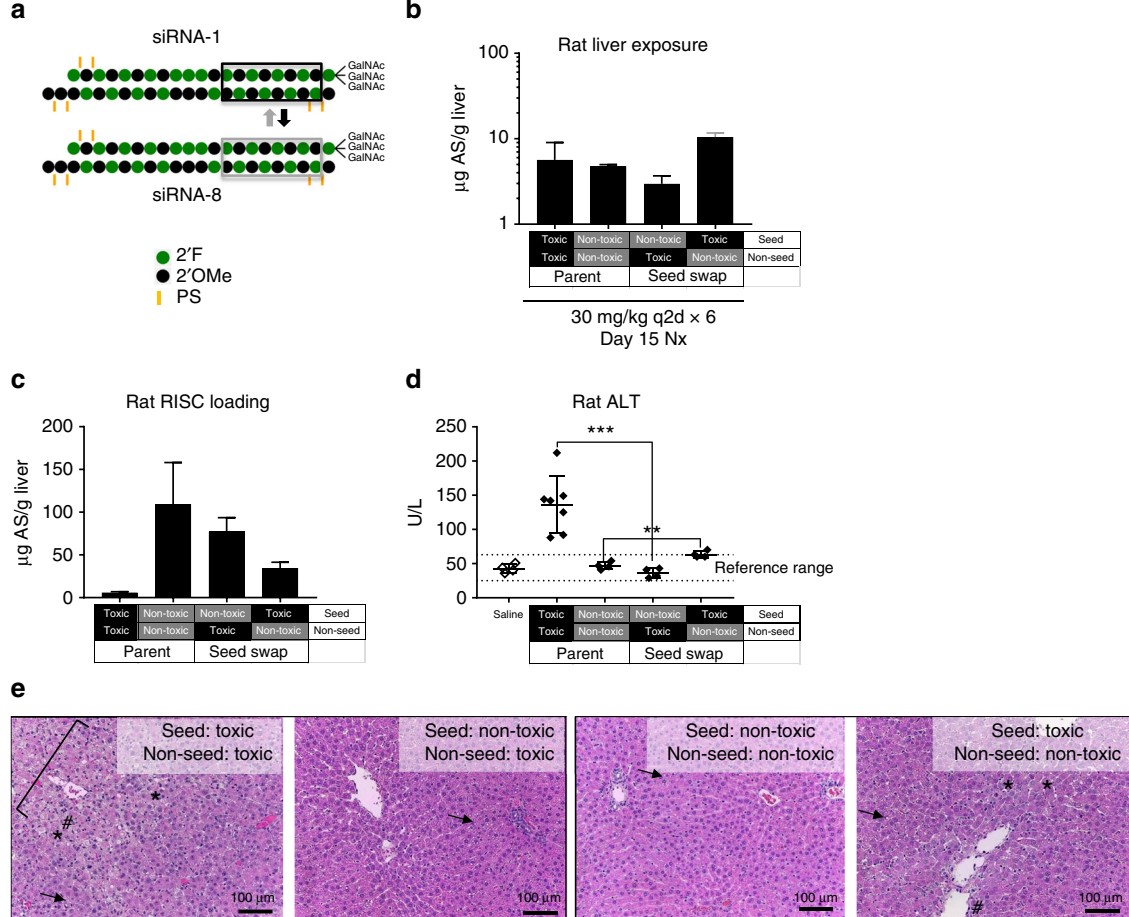

**Fig. 4** Swapping seed regions mitigates hepatotoxicity. **a** Chemical structures of seed swapping between a hepatotoxic and a non-hepatotoxic GalNAc-siRNA. **b** Liver exposures for parent and seed-swapped GalNAc-siRNAs in rat toxicity study as assessed by stem-loop RT-qPCR for the antisense strand (AS) at necropsy (nx). **c** Liver RISC loading as assessed by stem-loop RT-qPCR for the antisense strand at necropsy. **d** Serum alanine aminotransferase (ALT) levels measured at necropsy. Differences between group means were evaluated for statistical significance using one way ANOVA with post hoc corrections (for multiple siRNAs) in GraphPad Prism 7. ns, not significant; $**p < 0.01$, $***p < 0.001$. Error bars represent standard deviation of the mean. **e** H&E staining of liver sections collected at necropsy. The toxic siRNA had hepatocellular degeneration (bracket), single cell necrosis (*), increased sinusoidal cells consistent with Kupffer cell hyperplasia and/or infiltrating leukocytes (#), and hepatocellular vacuolation (arrow), while the non-toxic siRNA had only minimal vacuolation. The non-toxic seed in the toxic backbone was comparable to the full non-toxic siRNA, and the toxic seed in the non-toxic backbone had single cell necrosis, increased sinusoidal cells and vacuolation but at a lower severity grade than the full-length toxic compound. Microscopic liver findings for all tested siRNAs are tabulated in Supplementary Table 6. $N = 3$ males (6–8 weeks old) per group; q2d, every other day dosing

with a non-toxic seed region mitigated liver enzyme elevations (Fig. 4d) and microscopic liver findings (Fig. 4e; Supplementary Table 6), indicating that the seed region is necessary for hepatotoxicity with little to no contribution from siRNA chemistry. On the other hand, replacing a non-toxic seed region with a toxic seed region did not fully recapitulate hepatotoxicity of the toxic siRNA but did cause an increase in liver enzymes (Fig. 4d) and an increased severity of microscopic liver findings relative to the non-toxic parent siRNA (Fig. 4e and Supplementary Table 4). This suggests that while complementarity to the antisense seed region is required for off-target activity, the siRNA 3′ region may also contribute to off-target binding and repression. These data provide further support for RNAi-mediated, seed-based off-target effects as the major driver of rat hepatotoxicity.

**siRNA off-targets are enriched for seed complementarity**. To confirm that GalNAc-siRNAs can cause gene dysregulation consistent with RNAi-mediated off-target effects, a series of

siRNAs was transfected into rat hepatocytes to evaluate the global impact on the transcriptome by RNA sequencing (RNAseq) at a "toxicological" dose that exceeded the $IC_{50}$ concentration by 2–3 logs. Downregulated transcripts were enriched for perfect complementarity to the antisense seed region (nucleotides 2–8), and the magnitude of change generally did not exceed twofold (Fig. 5a). No such pattern of enrichment was observed for upregulated transcripts, or for the seed region of the sense strand. Similar off-target profile characteristics were observed in rat livers in vivo at 24 h following a 50 mg/kg dose of GalNAc-siRNA (Fig. 5b). The number of dysregulated genes was reduced with siRNAs containing 5′-end caps, indicating that the 2′F, 2′OMe, or PS chemistry and/or other RISC-independent factors do not significantly contribute to gene dysregulation, consistent with the results from rodent toxicity studies (Fig. 1). These data further support the conclusion that miRNA-like activity of the antisense strand[20–22], and not RNAi-independent effects based on siRNA chemistry, is the primary driver of off-target gene expression changes.

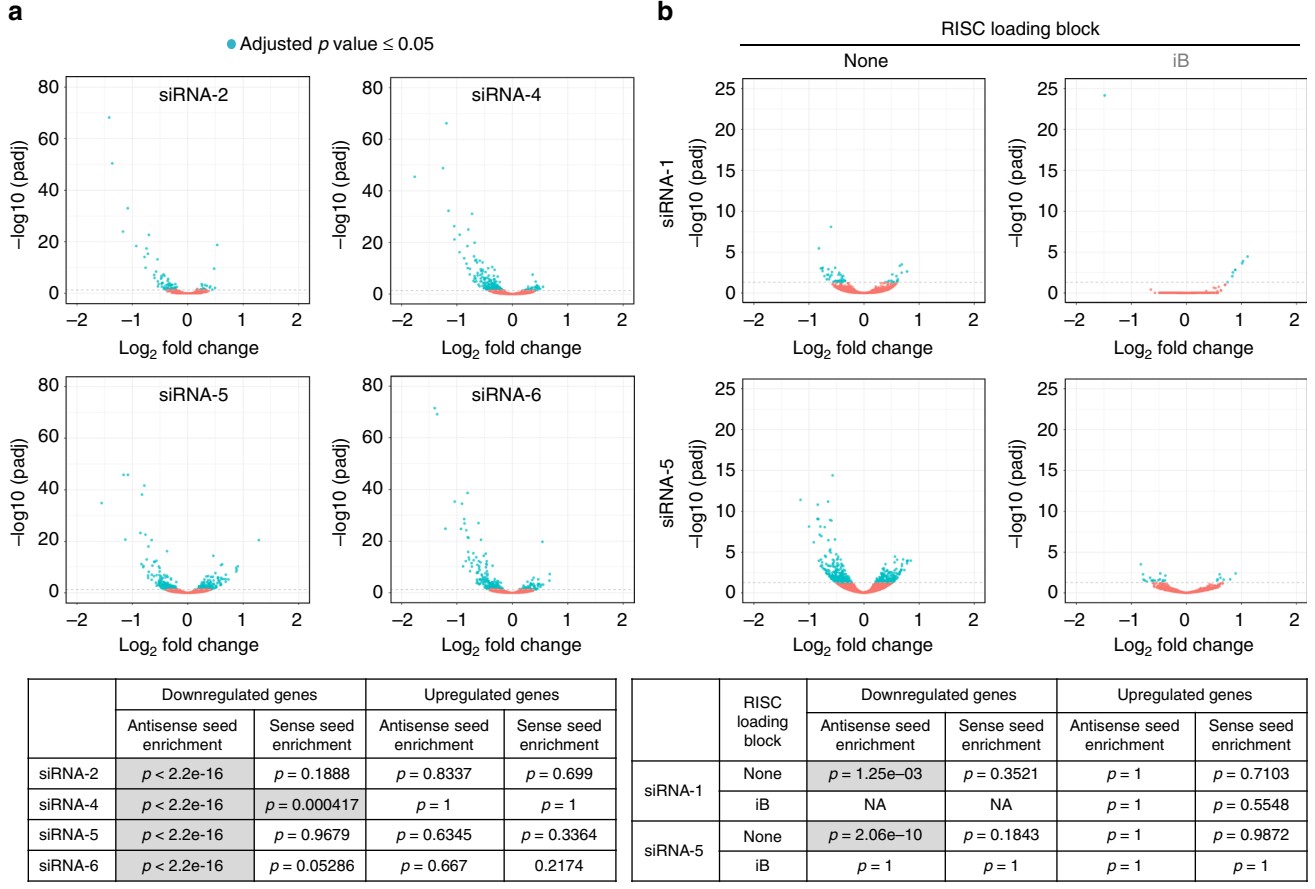

**Fig. 5** siRNA off-targets are enriched for seed complementarity in vitro and in vivo. **a** Volcano plots depicting global gene expression changes in rat hepatocytes at 24 h after transfection with 10 nM of GalNAc-siRNAs of four different sequences. Enrichment of the seed region complementarity in 3′UTRs is tabulated. $N = 3$ technical replicates. **b** Volcano plots depicting global gene expression changes in rat liver at 24 h after subcutaneous administration of GalNAc-siRNAs at 50 mg/kg. Two parent GalNAc-siRNAs and their RNAi-inactive versions blocked with inverted abasic (iB) caps are shown. Enrichment of the seed region complementarity in 3′UTRs is tabulated. Blue points, adjusted $p \leq 0.05$; red points, adjusted $p > 0.05$; $N = 3$ males (6–8 weeks old) per group. The adjusted $p$-value for fold change was calculate in DESeq2 using the Wald test with multiple test correction. Seed enrichment $p$-value was calculated using the Fisher's exact test. The variance was similar between groups that were statistically compared. iB, inverted abasic; NA, not applicable ($p$-values could not be calculated due to absence of downregulated genes)

**Impact of destabilizing seed-mediated off-target binding**. If seed-mediated recognition is necessary for off-target-driven hepatotoxicity of GalNAc-siRNAs, decreasing the binding affinity of the seed region to off-target mRNAs should have a mitigating effect. To test this hypothesis, a thermally destabilizing GNA nucleotide[34] was placed at position seven of the antisense strand in the hepatotoxic siRNA-5 sequence (Fig. 6a), analogous to previous approaches with other thermally destabilizing modifications[35–39].

The incorporation of GNA in the antisense strand seed region reduced the off-target signature compared to the parent siRNA when transfected into rat hepatocytes at a high dose of 10 nM (Fig. 6b), while maintaining on-target activity (Supplementary Fig. 7a). To further test whether reduction in the off-target signature translates into improved safety in vivo, these same two siRNAs were tested in a rat toxicity study with 3 weekly doses of 30 mg/kg. Relative to the parent sequence, GNA nucleotide substitution in the seed region did not impact on-target mRNA knockdown (Supplementary Fig. 7b), liver exposure (Fig. 6c), or RISC loading (Fig. 6d). However, seed modification mitigated liver enzyme elevations (Fig. 6e) and microscopic liver findings (Fig. 6f; Supplementary Table 7). In addition to providing additional evidence for off-target effects as the major driver of hepatotoxicity, these data provide the first reported evidence that

thermal destabilization of seed-mediated binding is a viable strategy for the selective reduction of off-target repression and hepatotoxicity of siRNAs in vivo.

## Discussion

The aim of this work was to elucidate the mechanisms of hepatotoxicity observed with a subset of liver-targeted GalNAc-siRNAs in repeat-dose sub-acute (2–8 week) rodent toxicity studies. Multiple plausible hypotheses were investigated utilizing a diverse set of tools and approaches. These included investigating potential mechanisms upstream of RISC loading, such as chemical toxicity of GalNAc-siRNAs and/or their metabolites, as well as potential effects downstream of RISC loading, which include competition with endogenous RNAi pathways and RNAi-based off-target repression of partially complementary mRNAs.

GalNAc-siRNAs described here are fully 2′OMe/2′F-modified at the 2′ position of the ribose and contain a limited number of terminal PS linkages[8, 11]. The accumulation of these stabilized oligonucleotides in the liver could potentially elicit hepatotoxicity by interfering with the endo-lysosomal system or intracellular protein binding, or by metabolite generation. The aggregate data presented herein provides multiple lines of evidence against the role of such mechanisms in the hepatotoxicity observed for a

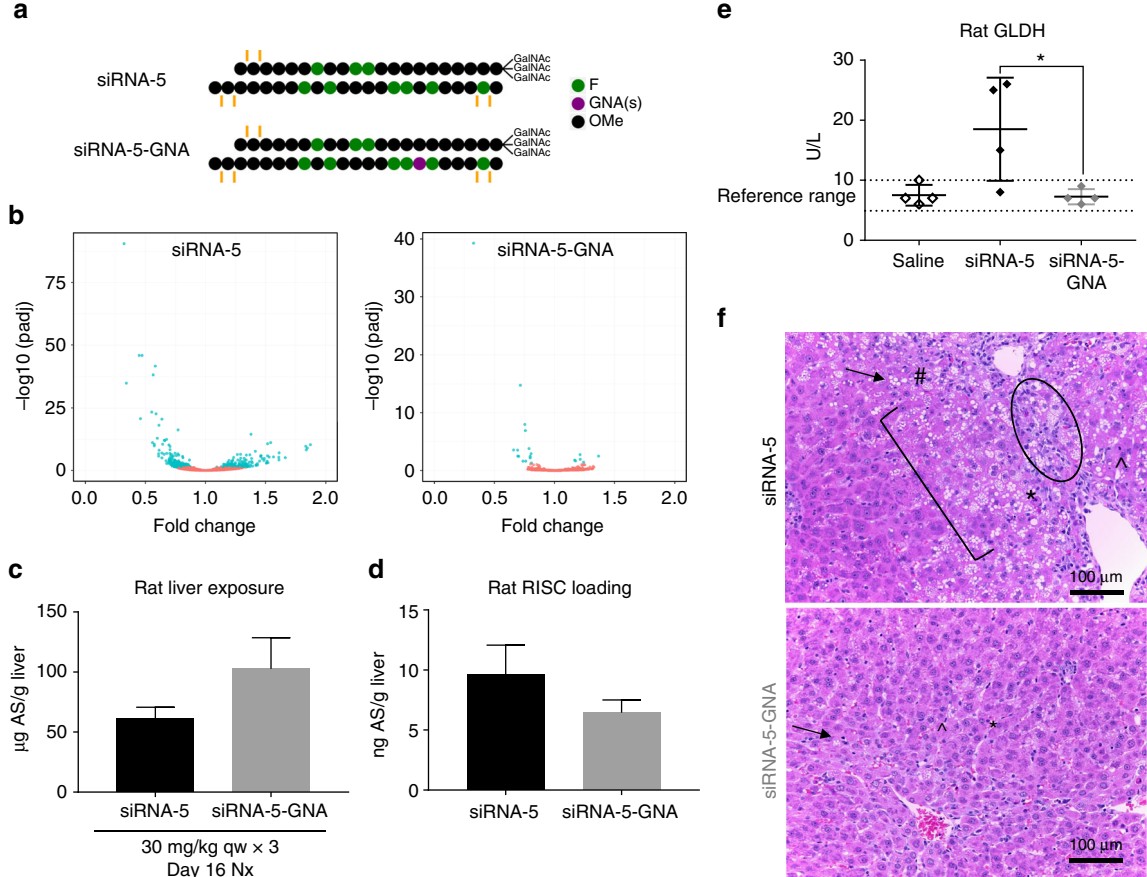

**Fig. 6** Destabilizing seed-mediated base-pairing minimizes off-target effects and mitigates hepatotoxicity. **a** Bad actor siRNA-5 containing a single thermally destabilizing glycol nucleic acid (GNA) nucleotide at position seven of the antisense strand. **b** Volcano plots depicting global gene expression changes in rat hepatocytes at 24 h after transfection with 10 nM of parent or GNA-modified GalNAc-siRNAs. $N = 3$ technical replicates. **c** Liver exposures for parent and seed-modified siRNA-5 in rat toxicity study as assessed by stem-loop reverse transcription-quantitative PCR (RT-qPCR) for the antisense strand (AS) at necropsy (nx). **d** Liver RISC loading as assessed by stem-loop RT-qPCR for the antisense strand at necropsy. **e** Serum glutamate dehydrogenase (GLDH) levels measured at necropsy. Differences between group means were evaluated for statistical significance using one way ANOVA with post hoc corrections (for multiple siRNAs) in GraphPad Prism 7. ns not significant; $*p < 0.05$, $**p < 0.01$, $***p < 0.001$, $****p < 0.0001$. Error bars represent standard deviation of the mean. **f** H&E staining of liver sections collected at necropsy. The toxic parent siRNA-5 had fibrosis (circle), hepatocellular degeneration (bracket), single cell necrosis (*), increased mitoses (^), increased sinusoidal cells consistent with Kupffer cell hyperplasia and/or infiltrating leukocytes (#), and hepatocellular vacuolation (arrow), while the non-toxic siRNA had only minimal vacuolation. The GNA-modified siRNA-5 had degeneration, single cell necrosis, increased mitoses, and vacuolation but at a lower incidence and severity grade than the parent siRNA-5. Microscopic liver findings for all tested siRNAs are tabulated in Supplementary Table 7. $N = 4$ males (6–8 weeks old) per group; qw weekly dosing, GNA glycol nucleic acid

random subset of hepatotoxic GalNAc-siRNAs in rodents. Specifically, the following approaches mitigated the hepatotoxicity observed with known toxic siRNAs at pharmacologically exaggerated doses, without changing the 2′F, 2′OMe, and PS content, or liver exposure: (1) blocking siRNA RISC loading with 5′ caps on the antisense but not the sense strand, (2) blocking off-target binding of siRNA–RISC complexes with a REVERSIR™, (3) swapping seed regions of known toxic and non-toxic siRNAs, and (4) utilizing seed-pairing destabilization with GNA in the antisense strand. In contrast, altering siRNA chemistry by partially substituting 2′F with 2′OMe modifications had no effect on hepatotoxicity.

In toxicity studies where liver exposures can exceed pharmacologically relevant exposures by 2–3 logs, it is conceivable that natural miRNAs could be displaced from RISC. However, blocking the activity of siRNA–RISC complexes with REVERSIR™ molecules and altering the sequence or binding affinity of antisense seed regions mitigated hepatotoxicity without reducing RISC loading, hence arguing against perturbation of natural

RNAi pathways as the cause of hepatotoxicity in rodents, as previously reported[27]. This suggests that the rate of continuously produced Ago proteins is high enough to maintain endogenous miRNA activity even in the presence of large quantities of exogenous siRNAs. In this respect, it is also conceivable that exogenous siRNA may load into and stabilize Ago proteins that would otherwise be degraded in the absence of sufficient small RNAs, a hypothesis that warrants further investigation.

Taken together, the evidence that we have presented points towards RNAi-mediated, hybridization-based off-target effects driven by the seed region of the antisense strand as the major cause for the rat hepatotoxicity observed with a random subset of hepatotoxic GalNAc-siRNA conjugates. This is further supported by our studies showing that transcripts dysregulated after siRNA treatment, observed by analyzing global effects on the transcriptome by RNAseq, are enriched for perfect complementarity to the antisense seed region.

While off-target effects of siRNAs have been observed before[20–22], this is, to our knowledge, the first report in which

hepatotoxicity findings in preclinical animal models could be linked to a specific RNAi-based mechanism. These mechanistic insights may guide strategies to reduce attrition rates in toxicity screens and to further enhance the safety of GalNAc-siRNAs by minimizing miRNA-like off-target effects. Indeed, selective mitigation of seed-mediated off-target binding can be achieved by introducing a single thermally destabilizing GNA nucleotide into the antisense seed region, resulting in an improved safety profile of a GalNAc-conjugated siRNA in the rat. Additional studies will be needed to evaluate the generalizability of thermally destabilizing modifications across sequences and positions, as well as the translation of this approach into higher species.

## Methods

**Care and use of laboratory animals.** All studies were conducted using protocols consistent with local, state and federal regulations, as applicable, and were approved by the Institutional Animal Care and Use Committee (IACUC) at Alnylam Pharmaceuticals. The test articles were diluted with 0.9% NaCl to achieve appropriate dosing concentrations and dosed subcutaneously on the upper back to male Sprague Dawley rats (6–8 weeks old) or male CD-1 mice (6–8 weeks old) in a dose volume of 5 mL/kg with $N = 3$ animals/group. Randomization was performed using the partitioning algorithm in the Pristima® Suite (Xybion) that avoids group mean body weight bias. Investigators were not blinded to the group allocation during the experiment or when assessing the outcome.

**Clinical pathology.** Whole-venous blood was collected into serum separator tubes (BD Microtainer) and allowed to clot at room temperature for 30 min prior to centrifugation at 3000 RPM ($1489 \times g$) for 10 min at 4 °C. Serum was then aliquoted and stored at −80 °C until analyses. Serum chemistries were analyzed using the AU400 chemistry analyzer (Beckman Coulter- Brea, CA, USA), with reagents provided by Beckman Coulter, Randox, and Sekisui Diagnostics. Differences between group means were evaluated for statistical significance using one-way ANOVA in GraphPad Prism 7.

**Histopathology.** All animals were killed as per Alnylam standard operating procedures and tissues of interest were collected. All tissues were fixed in 10% neutral buffered formalin (10% NBF) for 72 h prior to routine processing using TissueTek VIP 6A1 (Sakura). Tissues were trimmed, embedded into paraffin blocks, sectioned at 4 μ, stained with Hematoxylin and Eosin (H&E) using TissueTek Prisma A1D (Sakura), and coverslipped using TissueTek Glass g2 (Sakura). Two sections were examined microscopically from each liver in an un-blinded fashion, followed by blinded assessment to confirm subtle findings. The range of severity grade for each histologic finding was graded on a scale of 1–5 with 1 indicating minimal severity and 5 indicating severe severity.

**Monomer and oligonucleotide synthesis.** All oligonucleotides were synthesized on an ABI 394, MerMade 192, MerMade 12, or ÄKTA oligopilot 100 synthesizer[8, 34]. Phosphoramidite monomers of 2′-F (Hongene Biotechnology), 2′-OMe (Hongene Biotechnology), LNA (Exiqon), or inverted abasic (iB, Chem-Genes) were obtained from commercial sources. The synthesis of GNA phosphoramidites monomers has been previously reported[34]. 5′-deoxy-5′-(4-morpholinyl)-uridine, 5′-deoxy-5′-(4-morpholinyl)-cytidine, and 5′-deoxyuridine phosphoramidites were synthesized in-house and will be reported separately (Parmar et al., manuscript in preparation). All phosphoramidites were used at a concentration of 100 or 150 mM in 100% acetonitrile, 9:1 acetonitrile:DMF, or 85:15 acetonitrile:THF. Standard protocols for 2-cyanoethyl phosphoramidites utilizing ETT activator were used with coupling times of 6–7 min. Oxidation of phosphite linkages was achieved using $I_2$ in 9:1 acetonitrile:water, DDTT (3-(dimethylaminomethylene) amino-3H-1,2,4-dithiazole-5-thione; Rasayan) in 9:1 pyridine:acetonitrile, or PADS (phenylacetyl disulfide; American International Chemical) in 1:1 2,6-lutidine:acetonitrile. After the trityl-Off synthesis, cleavage from the support with subsequent removal of all protecting groups was accomplished using either 40% aqueous methylamine or 3:1 [$NH_4OH$]:EtOH at RT or 65 °C, respectively. Crude oligonucleotides were purified using strong anion exchange with phosphate buffers (pH = 8.5 or 11) containing NaBr, the appropriate fractions were pooled, and finally desalted. The identities and purities of all oligonucleotides were confirmed using ESI-LC/MS and IEX HPLC, respectively.

**Quantification of liver and Ago2-associated siRNA levels.** Liver and Ago2-associated (RISC-loaded) siRNA levels were quantified by stem-loop reverse transcription-quantitative PCR (RT-qPCR) as previously described[30, 40]. Total siRNA liver levels were measured by reconstituting liver powder at 10 mg/ml in PBS containing 0.25% Triton-X 100. The tissue suspension was further ground with 5-mm steel grinding balls at 50 cycles/s for 5 min in a tissue homogenizer (Qiagen TissueLyser LT) at 4 °C. Homogenized samples were then heated at 95 °C for 5 min, briefly vortexed and allowed to rest on ice for 5 min. Samples were then centrifuged at $21,000 \times g$ for 5 min at 4 °C. The siRNA-containing supernatants were quantified by stem loop reverse transcription followed by Taqman PCR.

Ago2-bound siRNA was quantified by preparing liver powder lysates at 100 mg/ml in lysis buffer (50 mM Tris–HCl, pH 7.5, 150 mM NaCl, 2 mM EDTA, 0.5% Triton-X 100) supplemented with freshly added protease inhibitors (Sigma-Aldrich, P8340) at 1:100 dilution and 1 mM PMSF (Life Technologies). Total liver lysate (10 mg) was used for each Ago2 immunoprecipitation (IP) and control IP. Anti-Ago2 antibody was purchased from Wako Chemicals (Clone No.: 2D4). Control mouse IgG was from Santa Cruz Biotechnology (sc-2025). Protein G Dynabeads (Life Technologies) were used to precipitate antibodies. Ago2-associated siRNAs were eluted by heating (50 μl PBS, 0.25% Triton; 95 °C, 5 min) and quantified by stem loop reverse transcription followed by Taqman PCR.

**RNAseq and bioinformatic analysis.** Rat livers were collected 24 h post-50 mg/kg single dose of GalNAc-siRNAs and snap-frozen. Rat hepatocytes (BioreclamationIVT) were transfected with 10 nM GalNAc-siRNAs using Lipofectamine RNAiMAX (Thermo Fisher Scientific) according to manufacturer's instructions, and harvested 24 h post-transfection. Rat hepatocytes were not tested for mycoplasma contamination. RNA extracted with the miRNeasy kit (Qiagen) was used for cDNA library preparation with the TruSeq Stranded Total RNA Library Prep Kit (Illumina) and sequenced on the HiSeq or NextSeq500 sequencers (Illumina), all according to manufacturers' instructions. Raw RNAseq reads were filtered with minimal mean quality scores of 25 and minimal remaining length of 36, using fastq-mcf. Filtered reads were aligned to the *Rattus norvegicus* genome (Rnor_6.0) using STAR (ultrafast universal RNAseq aligner) with default parameters. Uniquely aligned reads were counted by featureCounts[41]. Differential gene expression analysis was performed by the R package DESeq2[42]. The open source DESeq2 R package was used for the RNAseq data analysis.

**Data availability.** The data discussed in this publication have been deposited in NCBI's Gene Expression Omnibus[43] and are accessible through GEO Series accession number GSE108823 (https://www.ncbi.nlm.nih.gov/geo/query/acc.cgi?acc=GSE108823).

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

## Acknowledgments

We thank Anna Bisbe, Klaus Charisse, Ryan Malone, Jonathan O'Shea, Nate Taneja, and Christopher Theile for oligonucleotide synthesis; Terence Cawley, Jeff Rollins, Stacy Seide, and Scott Waldron for duplex annealing and characterization; Antoinette Hayes, Emma Henchy, Brian LeMay, Lauren Moran, Elena Ooms, Kaleigh Pavlik, and Catrina Wong for assistance with in vivo studies; Kristina Perry and Kellie Sawyer for clinical pathology analysis; Brenda Carito and Paul Gedman for histology slide preparation; Joseph Barry and Greg Hinkle for assistance with RNAseq data deposition.

## Author contributions

M.M.J, M.K.S., M.A.M., and V.J. designed the studies and analyzed the data. C.E.H. and N.D.K interpreted clinical pathology and histopathology data. V.O.Y. generated siRNA liver exposure and RISC loading data. Y.J. performed rat hepatocyte transfections and REVERSIR™ screening. M.K.S., I.Z., A.C., K.G.R., and M.A.M. designed and synthesized GalNAc-siRNA and REVERSIR™ compounds. R.P., K.G.R., and M.M. designed and synthesized RISC-loading blocking modifications. V.O.Y. and S.S.-M. generated and H.X. analyzed the RNA sequencing data. M.M.J., M.K.S., M.A.M., and V.J. wrote the manuscript with input from all authors.

## Additional information

**Competing interests:** All authors are employees of Alnylam Pharmaceuticals with salary and stock options. REVERSIR is a trademark of Alnylam Pharmaceuticals.

