## [Peer Review File · Nature Communications]

Editorial Note: This manuscript has been previously reviewed at another journal that is not operating a transparent peer review scheme. This document only contains reviewer comments and rebuttal letters for versions considered at Nature Communications. Mentions of prior referee reports have been redacted.

Reviewers' comments:

Reviewer #1 (Remarks to the Author):

This manuscript reported that high dose of GalNAc-siRNA induced toxicity through off-target effect of RNAi without effects of nucleic acids modifications. The results were proved by using a variety of new methodologies such as 5'phospholilation inhibition, REVERSIR, Seed swapping and summarized systematically. I believe that this study will give a significant impact on molecular design of siRNA based therapy for clinical application, which matches the quality of Nature Communications. However, following points should be carefully considered before this manuscript has been accepted for publication by Nature Communication.

- 1) Authors are advised to describe target genes for siRNA1 – 7.
- 2) Authors are advised to describe capping structure of RISC loading block in the Figure.
- 3) Authors should apply statistical analysis and discuss them in discussion.

Reviewer #2 (Remarks to the Author):

The authors investigate the mechanism GalNAc-modified siRNA induced toxicities in the livers of animals. They conclude that most of the toxicity is the result of RNAi-mediated off targeting with little contribution from the modifications. The fact is that the doses used were supra physiological and the mechanism other than off-targeting effects remain vague.

Biological and technical concerns.

- 1) What were the targets of the various siRNAs used in the study? Statements like dosing representing 2-3 log exaggeration of the pharmacological dose without knowing the target sequence and a dose response is not extremely helpful. A dose response for the targeted sequence should be provided.
- 2) Page 11- The compounds..... retained potent silencing activity. . Where is the data showing silencing. No such data is provided but should be included.
- 3) The details, quantification and statistical analyses of the histological results need to be provided. How were the scores in the table generated? How many sections from each mouse and was the examiner blinded to the treatment groups?
- 4) When providing Reversir data-- can the authors provide data that show the siRNA is functionally inhibited when the antisense oligo is given? For example take a bona-fide mRNA target and determine the degree of mRNA knockdown in the presence or absence of the antisense oligo.
- 5) The same type of analysis should be done for at least one target mRNA after delivery of RISC loaded and non-loaded siRNAs.
- 6) Bottom Page 13. Statements like RISC loading was lower for the toxic.... as well as ... need a comparator . Lower than what specifically?
- 6) The finding that the seed match is required but not sufficient for hepatic toxicity requires additional study (see the additional points below).
- 7) What is the target sequence of the toxic siRNAs and could the knockdown of the target be what induces toxicity? Moreover, in the bioinformatic study of the off-targeting was there differential changes in gene expression profiles between the toxic and non-toxic siRNAs and what about GO (or similar analyses) to try to understand the off-targeted genes that were responsible for

toxicity.

Reviewer #3 (Remarks to the Author):

This is an important study which highlights the role for seed pairing as a possible mechanism for toxicity. My only concern is that the authors did not propose or demonstrate a reasonable solution to avoid this problem. What is lacking in this manuscript is a proposed and validated mechanism to avoid such seed pairing. Perhaps a bio-informatics approach could be used to screen the transcriptome for possible siRNA seed pairing with all expressed mRNAs.

Reviewer #1 (Remarks to the Author):

This manuscript reported that high dose of GalNAc-siRNA induced toxicity through off-target effect of RNAi without effects of nucleic acids modifications. The results were proved by using a variety of new methodologies such as 5'phospholilation inhibition, REVERSIR, Seed swapping and summarized systematically. I believe that this study will give a significant impact on molecular design of siRNA based therapy for clinical application, which matches the quality of Nature Communications. However, following points should be carefully considered before this manuscript has been accepted for publication by Nature Communication.

The authors appreciate the overall positive response of the reviewer and the valuable suggestions, which have been made. We have addressed the specific points raised, as follows:

1) Authors are advised to describe target genes for siRNA1 – 7.

We added the target gene names and accession numbers for all siRNAs used in this manuscript to Supplementary Figure 1.

2) Authors are advised to describe capping structure of RISC loading block in the Figure.

We added the structures of the residues used to block the 5'-end and prevent phosphorylation that were utilized in this study and updated Figure 1 accordingly.

3) Authors should apply statistical analysis and discuss them in discussion.

Differences between group means were evaluated for statistical significance using one-way ANOVA in GraphPad Prism 7 for all clinical pathology data. p-values are included in all Figures, Figure Legends, and described in the "Clinical pathology" section of Materials and Methods on page 7.

Reviewer #2 (Remarks to the Author):

The authors investigate the mechanism GalNAc-modified siRNA induced toxicities in the livers of animals. They conclude that most of the toxicity is the result of RNAi-mediated off targeting with little contribution from the modifications. The fact is that the doses used were supra physiological and the mechanism other than off-targeting effects remain vague.

The authors appreciate the thorough review and the valuable suggestions for changes and additional data. We have addressed the specific points raised as follows:

Biological and technical concerns.

1) What were the targets of the various siRNAs used in the study? Statements like dosing representing 2-3 log exaggeration of the pharmacological dose without knowing the target

sequence and a dose response is not extremely helpful. A dose response for the targeted sequence should be provided.

We added target genes for all siRNAs used in this manuscript to Supplementary Figure 1. For those siRNAs that are cross-reactive to rodents, the ED50 dose is typically in the range of 0.05 - 0.5 mg/kg (see Supplementary Figure 5), while rat toxicity screening studies are typically dosed at 30-100 mg/kg.

2) Page 11- The compounds..... retained potent silencing activity. . Where is the data showing silencing. No such data is provided but should be included.

We added in vivo silencing data for the high 2'F and low 2'F siRNAs to new Supplementary Figure 5 showing equivalent potency.

3) The details, quantification and statistical analyses of the histological results need to be provided. How were the scores in the table generated? How many sections from each mouse and was the examiner blinded to the treatment groups?

We added the requested details on the histopathological evaluation to the "Histopathology" section of Materials and Methods on page 8.

4) When providing Reversir data-- can the authors provide data that show the siRNA is functionally inhibited when the antisense oligo is given? For example take a bona-fide mRNA target and determine the degree of mRNA knockdown in the presence or absence of the antisense oligo.

The effects of each Reversir on siRNA functionality are shown in Supplementary Figure 6.

5) The same type of analysis should be done for at least one target mRNA after delivery of RISC loaded and non-loaded siRNAs.

The effects of RISC loading blocks on siRNA functionality (for those siRNAs that were cross-reactive to rat) are shown in Supplementary Figure 2b.

6) Bottom Page 13. Statements like RISC loading was lower for the toxic.... as well as ... need a comparator . Lower than what specifically?

We added a comparator to clarify this sentence in the "Swapping seed regions mitigates hepatotoxicity" section on page 14.

6) The finding that the seed match is required but not sufficient for hepatic toxicity requires additional study (see the additional points below).

To further support this conclusion, we added a new section in the Results entitled "Impact of destabilizing seed-mediated off-target binding" on pages 15-16, and corresponding Figure 6, Supplementary Figure 7, and Supplementary Table 5. This data demonstrates that seed-pairing destabilization, utilizing a novel GNA nucleotide, can improve the safety profile of the example siRNA without impacting potency.

7) What is the target sequence of the toxic siRNAs and could the knockdown of the target be what induces toxicity? Moreover, in the bioinformatic study of the off-targeting was there differential changes in gene expression profiles between the toxic and non-toxic siRNAs and what about GO (or similar analyses) to try to understand the off-targeted genes that were responsible for toxicity.

Although not shown in the manuscript, for each target gene we were able to identify both non-toxic and toxic siRNA sequences, arguing against on-target toxicity. GO analyses of our limited RNAseq datasets to date did not reveal any signatures that can differentiate non-toxic and toxic siRNAs, but we are in the process of generating additional RNAseq data that may enable us to discern such patterns.

Reviewer #3 (Remarks to the Author):

This is an important study which highlights the role for seed pairing as a possible mechanism for toxicity. My only concern is that the authors did not propose or demonstrate a reasonable solution to avoid this problem. What is lacking in this manuscript is a proposed and validated mechanism to avoid such seed pairing. Perhaps a bio-informatics approach could be used to screen the transcriptome for possible siRNA seed pairing with all expressed mRNAs.

The authors appreciate overall positive review and the valuable feedback. We have addressed the reviewer's suggestion for demonstrating a reasonable solution to avoid off target effects as follows:

We added a new section in the Results entitled "Impact of destabilizing seed-mediated off-target binding" on pages 15-16, and corresponding Figure 6, Supplementary Figure 7, and Supplementary Table 5 that describe a potential mitigation strategy. This data demonstrates that seed-pairing destabilization, utilizing a novel GNA nucleotide, can improve the safety profile of the example siRNA without impacting potency.

We don't believe a bioinformatics approach is feasible to screen the transcriptome for possible siRNA seed pairing with all expressed mRNAs because a randomly-chosen heptamer seed will appear in an average of ~1,500 3'UTRs, with 95% of heptamers appearing in 25 to ~3,500 genes. Although we are actively trying to understand what defines a functional seed site to refine our off-target prediction algorithm, we believe mitigation is a better strategy than prediction.

Reviewers' Comments:

Reviewer #3:

Remarks to the Author:

The authors have made honest efforts to address the critiques of the reviewers.

Reviewer #4:

Remarks to the Author:

This manuscript by scientists from Alnylam describes novel insights into hepatotoxicity of siRNAs in rodents. The toxicity was found to be sequence specific and attributed to miRNA-seed region binding. The authors elegantly use subtle chemical variations to influence pharmacological and toxicological outcomes. The data are important findings for developing safe and effective oligonucleotide drugs and the article advances the field.

If adverse effects are sequence-dependent, it is certainly good news for further therapeutic development of GalNAc-siRNAs. However, it should be acknowledged that showing sequence-dependent hepatotoxicity in these preclinical models is not proof of absence of any class effect or other adverse effects that may be caused by the chemical modifications, and other mechanisms may be present in humans. This is shortly, but adequately discussed at the end of the manuscript.

The data are well and concisely presented and the text is written very clearly.

I was not a reviewer of the original submission, and per request of the editor I focused on the revision and responses made with regard to the comments of reviewer 2 on the first version. All of these comments and suggestions have been considered appropriately in the revised manuscript.

Specific points (numbering according to comments of reviewer 2):

- 1) and 2) Importantly, oligonucleotide sequences and targets as well as target knockdown of parent and modified (non-toxic) siRNAs were added.
- 3) Likewise, the requested details of histological grading were added. Investigators were generally not blinded to group allocation (apart from detailed histological grading). While it may be a certain limitation of the study, it is clearly stated in the manuscript and acceptable for preclinical rodent studies. Statistical testing method (one-way ANOVA) was added. This should be supplemented with post-hoc corrections (for multiple siRNAs).
- 4) and 5) The degree of functional inhibition by Reversir and by blocking RISC loading is depicted in the supplement.
- 6) The authors have provided new data for chemically destabilization of seed region affinity. The results show that off-targets are reduced and toxicity is mitigated and further substantiate the other outcomes.
- 7) Although not included in the article, the investigations of several siRNA against one target with only some of those causing hepatotoxicity is plausible and backed up by several literature reports. I suggest referencing this issue in the introduction; for example by stating that the 40 % toxic compounds in screening (p4) have little or no association with specific targets. See also comment below.

Other comments:

The selection of siRNA sequences that were tested is not completely clear: Are they a random selection of compounds for which hepatotoxicity was detected in screening? Or are these a subset of toxic oligonucleotides, namely those that share a common mechanism? In other words, do the authors know of siRNAs which exert sequence-independent toxicity in rodents? The sentence 'Here we describe a series of mechanistic studies demonstrating that RNAi-mediated, seed hybridization-based off-target effects are the major driver of hepatotoxicity observed with a subset of GalNAc-siRNAs in rodent toxicity studies.' (p 5 and similarly in the discussion) is slightly ambiguous with regard of 'subset'. Please clarify.

It would be interesting to know whether the adverse effects of revusiran encountered in clinical testing are linked to this mechanism, but I understand that this is beyond the scope of this article.

Minor point: It is good scientific practice to name commercial suppliers (of phosphoramidites, p8).

Very minor remark: consider rephrasing 'severe severity' on page 9 and Suppl. Tables.

NCOMMS-17-14859A

The authors appreciate the overall positive response of the reviewer and the valuable suggestions, which have been made.

Reviewer #4 (Remarks to the Author):

Responses in **red bold font**

This manuscript by scientists from Alnylam describes novel insights into hepatotoxicity of siRNAs in rodents. The toxicity was found to be sequence specific and attributed to miRNA-seed region binding. The authors elegantly use subtle chemical variations to influence pharmacological and toxicological outcomes. The data are important findings for developing safe and effective oligonucleotide drugs and the article advances the field.

If adverse effects are sequence-dependent, it is certainly good news for further therapeutic development of GalNAc-siRNAs. However, it should be acknowledged that showing sequence-dependent hepatotoxicity in these preclinical models is not proof of absence of any class effect or other adverse effects that may be caused by the chemical modifications, and other mechanisms may be present in humans. This is shortly, but adequately discussed at the end of the manuscript.

The data are well and concisely presented and the text is written very clearly.

I was not a reviewer of the original submission, and per request of the editor I focused on the revision and responses made with regard to the comments of reviewer 2 on the first version. All of these comments and suggestions have been considered appropriately in the revised manuscript.

Specific points (numbering according to comments of reviewer 2):

- 1) and 2) Importantly, oligonucleotide sequences and targets as well as target knockdown of parent and modified (non-toxic) siRNAs were added.
- 3) Likewise, the requested details of histological grading were added. Investigators were generally not blinded to group allocation (apart from detailed histological grading). While it may be a certain limitation of the study, it is clearly stated in the manuscript and acceptable for preclinical rodent studies. Statistical testing method (one-way ANOVA) was added. This should be supplemented with post-hoc corrections (for multiple siRNAs).

Response: Post-hoc correction for multiple siRNAs was included and is now specified in all relevant figure legends.

- 4) and 5) The degree of functional inhibition by Reversir and by blocking RISC loading is depicted in the supplement.
- 6) The authors have provided new data for chemically destabilization of seed region affinity. The results

show that off-targets are reduced and toxicity is mitigated and further substantiate the other outcomes. 7) Although not included in the article, the investigations of several siRNA against one target with only some of those causing hepatotoxicity is plausible and backed up by several literature reports. I suggest referencing this issue in the introduction; for example by stating that the 40 % toxic compounds in screening (p4) have little or no association with specific targets. See also comment below.

Response: We added this statement to the introduction as Track Changes on pg. 4.

Other comments:

The selection of siRNA sequences that were tested is not completely clear: Are they a random selection of compounds for which hepatotoxicity was detected in screening? Or are these a subset of toxic oligonucleotides, namely those that share a common mechanism? In other words, do the authors know of siRNAs which exert sequence-independent toxicity in rodents? The sentence 'Here we describe a series of mechanistic studies demonstrating that RNAi-mediated, seed hybridization-based off-target effects are the major driver of hepatotoxicity observed with a subset of GalNAc-siRNAs in rodent toxicity studies.' (p 5 and similarly in the discussion) is slightly ambiguous with regard of 'subset'. Please clarify.

Response: This subset is a random selection of compounds for which hepatotoxicity was detected in rat screening. We clarified this throughout the manuscript as Track Changes (pg. 5, 10, 17).

It would be interesting to know whether the adverse effects of revusiran encountered in clinical testing are linked to this mechanism, but I understand that this is beyond the scope of this article.

Minor point: It is good scientific practice to name commercial suppliers (of phosphoramidites, p8).

Response: As requested, we named all the commercial suppliers in methods section.

Very minor remark: consider rephrasing 'severe severity' on page 9 and Suppl. Tables.

Response: "Severe" is in the standard histopathology lexicon and we would prefer not to change this in the manuscript.